# Drawing Reliable Conclusions with Imperfect Synthetic Data

## Abstract

Predictions and generations from large language models are increasingly being explored as an aid in limited data regimes, such as in computational social science and human subjects research. While prior technical work has mainly explored the potential to use model-predicted labels for unlabeled data in a principled manner, there is increasing interest in using large language models to generate entirely new synthetic samples (e.g., synthetic simulations), such as in responses to surveys. However, it remains unclear by what means practitioners can use synthetic data alongside real data without invalidating downstream statistical conclusions. In this paper, we introduce a new estimator based on generalized method of moments, providing a hyperparameter-free solution with strong theoretical guarantees to address this challenge. We find that interactions between the moment residuals of synthetic data and those of real data (i.e., when they are predictive of each other) can substantially improve estimates of the target parameter. To the best of our knowledge, our framework provides the first theoretically-sound approach for incorporating fully synthetic samples in downstream statistical analyses.

## 1 Introduction

Practitioners increasingly leverage large language models (LLMs) as cheap but noisy labelers for automating tasks traditionally reliant on manual human annotations Ziems et al. [2024]. Beyond annotation, recently, practitioners have started to explore the possibility of leveraging LLMs for more diverse and open-ended forms of model-generated data. For instance, practitioners have started to leverage LLMs to output entirely new synthetic samples, e.g., simulating human responses to surveys or human participants in early pilot studies [Argyle et al., 2023, Brand et al., 2023, Dominguez-Olmedo et al., 2024, Anthis et al., 2025, Hwang et al., 2025b]. Determining the extent to which researchers should integrate LLM simulations—whether by simulating all samples, combining simulated and real samples, or relying entirely on human participants—remains an open and task-dependent question. While such pipelines leveraging fully synthetic simulations have yet to be fully realized, developing reliable mechanisms for aggregating these data sources is indeed what will inform both the feasibility of such design choices and how such pipelines should be implemented in practice.

A persistent challenge, however, is that naively aggregating synthetic samples with real data for downstream inference often lead to greatly biased estimates, compromising statistical validity of downstream conclusions. Ideally, we would like to realize the benefits of incorporating information from these additional data sources, while retaining favorable statistical properties—consistency and proper asymptotic coverage. We consider the setting where practitioners have access to a corpus of unlabeled text and a small set of human annotated samples with labeled covariates and outcomes. Here, practitioners can leverage LLMs to (1) predict covariates and outcomes for the unlabeled text

Submitted to 39th Conference on Neural Information Processing Systems (NeurIPS 2025). Do not distribute.

samples; and (2) generate new text samples conditioned on available samples and label the covariates and outcomes for them similarly to (1).

First of all, it is not immediately obvious how to even produce synthetic samples such that they can be used in a principled manner. Naively drawing samples from a generative model and treating them as additional samples alongside real data makes it impossible to provide statistical guarantees for the resulting estimate if the generative model does not perfectly match the real distribution—which is expected in practice. We propose a specific sampling strategy in which each synthetic sample is generated conditional on an individual real text as an example (see Section 3 for details). What makes this formulation statistically powerful is that it introduces a correlation structure between each real text and synthetic sample. This correlation structure will prove critical for principled methods for integrating synthetic data, as it enables us to more effectively share information across them.

We introduce a new estimation framework based on generalized method of moments (GMM) that naturally incorporates this synthetic data. The construction of our GMM estimator defines *separate* parameters and corresponding moments for each data source to avoid distributional assumptions (i.e., assuming that the generative model's distribution will match the real distribution). Prospectively, it is not intuitive that the incorporation of moments based exclusively on synthetic data (defined in terms of a separate parameter, independent of the target parameter) should yield any benefits (or *even affect*) the estimation of the target parameter. Intriguingly, we find that the interactions between the moment residuals of synthetic data and that of real data in the weighting matrix can substantially improve estimates of the target parameter (see Sections 4.5 and 6). The key intuition is that synthetic data will lead to more precise estimation and tighter confidence intervals when the synthetic data residuals are predictive of the real data residuals. When the synthetic residuals are independent of the observed data residuals, the variance reduces to the optimal variance based only on the fully observed data. That is, in the worst case where synthetic data is *completely uninformative*, including it does not hurt (at least asymptotically).

At a fundamental level, this work takes a first step towards understanding how imperfect synthetic data from foundation models can systematically be leveraged to support valid inference. As the usage and future promise of foundation models continue to grow, so too will the complexity of pipelines that incorporate their outputs. Our framework provides a foundation for easily extensible estimation methods that can safely incorporate the growing variety and quality of synthetic data sources from such models. More broadly, this GMM-based estimation framework for incorporating auxiliary data may be of broader interest as an alternative to the predominant debiasing-based methods in the surrogacy literature [Angelopoulos et al., 2023a], as it can more flexibly accommodate multiple proxy covariates and proxy outcomes compared to existing approaches.

## 2   Related Work

**LLMs for Data Annotation and Synthetic Simulation Tasks.**   Our work is motivated by the increasingly growing use and future promise of LLMs for annotations and simulation studies, particularly as a means to reduce human labeling costs [Hwang et al., 2025a]. Recently, LLMs have been tested in fully synthetic simulation studies [Dillion et al., 2023, Anthis et al., 2025], with primary applications in exploratory research or early pilot studies. For instance, recent work has studied simulating individuals in society and their interactions [Park et al., 2022, Chen et al.], analyzing whether the resulting LLM agents produced accurate responses on surveys and accurately predicted behavioral outcomes [Park et al., 2023]. Other works have applied LLMs to simulate survey responses [Geng et al., 2024, Rothschild et al., 2024], while others have cautioned about specific flaws in LLM responses [Dominguez-Olmedo et al., 2024], such as not accurately reflecting the influence of demographic groups [Dominguez-Olmedo et al., 2024, Wang et al., 2025]. In summary, this line of work shows the potential of fully synthetic experiments powered through strong generative models but also exhibits clear failure modes and imperfect conclusions from such studies. While most of these studies focus on qualitative takeaways and early signals for future experiments, we focus on the challenging and forward-looking setting of making statistically valid inference given such synthetic samples.

**Statistical Inference and Debiasing Methods.**   Our work is broadly related to performing statistical inference with missing data, where past works have explored approaches to yielding valid and efficient parameter estimates [Robins et al., 1994]. Other work has notably explored the usage of ML models

to estimate nuisance parameters [Chernozhukov et al., 2018]. The most related line of research are debiasing methods [Egami et al., 2023, Gligorić et al., 2024] that focus on combining ground truth data with surrogate predictions (often produced by a machine learning model) to perform statistical inference. These frameworks are often referred to as prediction-powered inference [Angelopoulos et al., 2023a,b] in the machine learning literature. A key difference between these works and our setting is that the primary focus of our work is how to incorporate fully synthetic samples, which remains unaddressed by previous work.

## 3 Preliminaries

**Notation and Setup.**   We consider a parameter estimation task where the goal is to estimate a target parameter $\theta^\star \in \mathbb{R}^d$. Let $(T, X, Y) \sim \mathcal{D}$ denote a random triple drawn from an unknown data-generating distribution $\mathcal{D}$ over text inputs $T \in \mathcal{T}$, covariates about the text (e.g., structured metadata) $X \in \mathcal{X} \subseteq \mathbb{R}^d$, and labels $Y \in \mathcal{Y}$. For example, $T$ can be texts from online requests, where $X$ are linguistic markers of hedging (i.e., notions of uncertainty) and $Y$ is perceived politeness. Due to labeling budget constraints, we assume we only observe a small fraction of human-annotated data (i.e., ground-truth covariates and labels about the text). Specifically, we have access to labeled dataset $\mathcal{D}_{\text{labeled}} = \{(T_i, X_i, Y_i)\}_{i=1}^n$ that is sampled i.i.d. from $\mathcal{D}$ and an unlabeled corpus of text $\mathcal{D}_{\text{unlabeled}} = \{(T_j)\}_{j=n+1}^{n+m}$ sampled i.i.d. from $\mathcal{D}_T$ (i.e., the marginal distribution over $T$), where $m \gg n$. To supplement this limited supervision, we leverage machine learning models in the following two ways.

**Proxy Covariates and Labels.**   We use a machine learning model $f$ to produce predictions $\{f_X(T_j), f_Y(T_j)\}$ for the available set of input texts $T \in \mathcal{T}$. Here, $f_X$ and $f_Y$ denote the same machine learning model, using separate prompts for the target outcome (either a covariate $X$ or outcome $Y$) (see Appendix for details). This yields the following $\mathcal{D}_{\text{proxy}} = \{(T_i, f_X(T_i), f_Y(T_i)\}_{i=1}^n \cup \{(T_j, f_X(T_j), f_Y(T_j)\}_{j=n+1}^{n+m}$. For simplicity, we will refer to this as **proxy samples** and denote them as $(T, \hat{X}, \hat{Y})$. We will refer to the distribution over proxy samples as $\hat{D}$. Note that this is the setting previous works have considered (mainly restricted to predicted outcomes) when addressing this problem.

**Synthetic Covariates and Labels.**   We propose a new data augmentation process which generates new samples using a text-based foundation model (employing it as a generative model, instead of a classifier as in previous works studying the proxy setup). Specifically, our method conditions the generation process on each individual text $T_j$ as an example and asks the model to generate a new synthetic sample given that context. Formally, for each $i$, we sample a new text $\tilde{T}_i$, conditioned on $(T_i, X_i)$ if the sample is labeled and $(T_j, \hat{X}_j)$ if the sample is unlabeled. For example, "`Consider text taken from user requests on Stack Exchange, either containing a hedging device or not containing one. {Insert example` $T_i$ `and covariate` $X_i$`}. Now, generate a new example of a request that matches the style of the provided example.`"[1]   Based on the generated sample, which we denote as $\tilde{T}_i$, we then extract its corresponding covariates and outcomes similarly as in proxy samples. More concretely,

$$\tilde{T}_k \sim \mathbb{P}(\cdot \mid T_i, X_i) \text{ if labeled,} \qquad \tilde{X}_k \sim \mathbb{P}(\cdot \mid \tilde{T}_k),$$
$$\tilde{T}_k \sim \mathbb{P}(\cdot \mid T_j, \hat{X}_j) \text{ if unlabeled} \qquad \tilde{Y}_k \sim \mathbb{P}(\cdot \mid \tilde{T}_k)$$

resulting in the following $\mathcal{D}_{\text{synthetic}} = \{(\tilde{T}_k, \tilde{X}_k, \tilde{Y}_k)\}_{k=1}^{n+m}$. We will refer to the distribution over **synthetic samples** $(\tilde{T}, \tilde{X}, \tilde{Y})$ as $\tilde{D}$.

This specific sampling process has two motivations. First, from a machine learning perspective it can be seen as a form of in-context prompting, where the model is given an example from the dataset in order to align it more closely with the task. Iteratively prompting with different samples $T_i$ is also likely to produce more diverse samples than asking for many samples with the same prompt. Second, from a statistical perspective, it introduces a correlation structure between each real text $T_i$ and synthetic sample $\tilde{T}_i$. This correlation structure will prove critical for principled methods for integrating synthetic data because it allows us to more effectively share information across them.

---

[1]See Appendix for further prompt details.

138 Indeed, naively drawing a set of synthetic samples from the generative model and pooling them with
139 the real data would render it impossible to provide statistical guarantees for the resulting estimate if
140 generative model fails to perfectly match the real distribution.

141 Finally, we introduce some notation that combines all of these data sources into draws from a single
142 joint distribution. Specifically, we introduce a new random variable $s \in \{0, 1\}$ which is an indicator
143 for whether $T$ is labeled (1) or unlabeled (0). Then, we view the complete generative process as
144 draws $(T, s, s \cdot X, s \cdot Y, \tilde{X}^1, \tilde{Y}^1 ... \tilde{X}^M, \tilde{Y}^M)$ for $M$ different kinds of auxiliary data. So far, we have
145 discussed two kinds, proxy and synthetic, that we employ empirically ($M = 2$), but our methods
146 are fully extensible to additional kinds of auxiliary data. For example, we could include samples
147 from multiple different generative models. The real $(X, Y)$ are observed only for labeled points with
148 $s = 1$ while the auxiliary data is available for all samples. The joint distribution over this full tuple is
149 induced by the composition of the generative processes for the components described above.

## 4   Combining Synthetic Information via Generalized Method of Moments

151 To estimate the target parameter $\theta^\star$, we adopt a generalized method of moments (GMM) approach
152 [Hansen, 1982] that combines information from the different types of data in the following manner.

### 4.1   Moment Conditions

154 Our framework is applicable whenever the target parameter can be identified by a set of moment
155 conditions, functions whose expectation should be zero at the true value of the parameter. Moment-
156 based estimation is a broad and flexible framework that includes almost all commonly used statistical
157 frameworks (e.g., maximum likelihood, generalized linear models, instrumental variables, etc). We
158 begin by defining the moment conditions that identify $\theta^*$ under the distribution of interest (i.e.,
159 the real-data distribution $\mathcal{D}$). In the following section, we introduce how this can be adapted to
160 incorporate surrogate data (i.e., proxy and synthetic data).

161 Formally, we consider the problem of estimating a parameter $\theta \in \mathbb{R}^d$. The true value $\theta^*$ is identified
162 as the solution to a set of $p \geq d$ moment conditions

$$\mathbb{E}[\psi^{(\ell)}(\theta^*)] = 0, \quad \ell = 1...p$$

163 where the $\psi^{(\ell)}$ are continuously differentiable functions $\mathbb{R}^d \to \mathbb{R}$. For example, in a maximum
164 likelihood model, we would have one $\psi$ for the derivative of the log-likelihood with respect to
165 each parameter, and the moment conditions enforce that $\theta^*$ satisfies the first-order conditions for
166 maximizing the likelihood. Let $\psi(\theta) = [\psi^{(1)}(\theta)...\psi^{(p)}(\theta)]^\top$ denote a column vector stacking the $p$
167 moments.

### 4.2   Constructing Our GMM Estimator

169 To leverage the auxiliary data (i.e., proxy data and synthetic data) in making our GMM estimator
170 more efficient, we can construct a set of auxiliary moments for each additional source of data. We
171 estimate an additional set of auxiliary parameters $\eta_1, ..., \eta_M \in \mathbb{R}^p$, one parameter vector for each
172 set of new auxiliary data. In the specific instantiation of the model that we use here, we always
173 have $M = 2$ (proxy and synthetic data), but in principle our method is extensible to many sources
174 of auxiliary data, for example synthetic samples generated from several different models. Roughly,
175 each new parameter vector $\eta_i$ can be understood as the parameter that we would estimate using
176 each auxiliary data source, and our augmented model will automatically determine how to use these
177 auxiliary estimates to inform the estimate of the parameter of interest $\theta$.

178 For each new parameter vector $\eta_i$, we introduce a corresponding set of new moments to estimate
179 this parameter and allow its estimate to inform the estimate of $\theta$. Specifically, we introduce for each
180 $\eta_i$ two new blocks of moments that are copies of the original moments for $\theta$. Intuitively, one block
181 of moments will be evaluated only on the real (labeled) data, while the other will be taken on the
182 pooled set of labeled data and auxiliary dataset $i$. The pooled-data moment will allow us to improve
183 the estimation of $\eta_i$ using the larger sample. The version evaluated only on the real data will allow
184 the GMM to evaluate how well the moments for the auxiliary parameter correlate with those of the
185 true parameter on the same data, and share information across them if the auxiliary moments are
186 informative (as we would expect if the generated data is high quality).

187  Formally, let $S_t \in \mathbb{R}^p$ stack $p$ copies of the indicator variable $s_t$ for whether a data point $t$ is labeled.
188  In block matrix notation, the combined model takes the form of the augmented moments

$$
g_t(\theta, \eta) =
\begin{bmatrix}
S_t \\
S_t \\
\vdots \\
S_t \\
1 \\
\vdots \\
1
\end{bmatrix}
\odot
\begin{bmatrix}
\psi(\theta) \\
\psi(\eta_1) \\
\vdots \\
\psi(\eta_M) \\
\psi(\eta_1) \\
\vdots \\
\psi(\eta_M)
\end{bmatrix}
\in \mathbb{R}^{p+2Mp}
\tag{1}
$$

189  We will then jointly estimate $(\theta, \eta)$ as the solution to the moment condition $\mathbb{E}[g_t(\theta, \eta)] = 0$. For
190  clarity, we refer to our estimator that uses real and proxy data ($M = 1$) as **GMM-Proxy** and our
191  estimator that uses real, proxy, and synthetic data ($M = 2$) as **GMM-Synth** throughout the paper.
192  We remark that since the parameter of interest $\theta$ appears only in its original set of moments, which
193  are evaluated only on the labeled data, this new moment condition still identifies the target parameter
194  $\theta^*$. However, as we discuss below, when we apply standard methods for efficiently estimating the
195  augmented GMM, the new moment conditions will be leveraged to reduce the variance of the estimate
196  without compromising consistency or asymptotic normality.

### 4.3  GMM Estimation

198  Given our augmented moment conditions $g$, we estimate the parameters $(\theta, \eta)$ by minimizing the
199  GMM objective:

$$
\hat{\theta}_T, \hat{\eta}_T = \arg \min_{\theta \in \Theta, \eta \in \mathbb{R}^{2Mp}} \widehat{Q}_T(\theta, \eta),
\tag{2}
$$

200  where

$$
\widehat{Q}_T(\theta, \eta) = \left[ \frac{1}{T} \sum_{t=1}^{T} g_t(\theta, \eta) \right]^{\top} \widehat{\mathbf{W}}_T \left[ \frac{1}{T} \sum_{t=1}^{T} g_t(\theta, \eta) \right].
\tag{3}
$$

201  Here, $\widehat{\mathbf{W}}_T \in \mathbb{R}^{M \times M}$ is a (possibly data-dependent) positive semidefinite weighting matrix that
202  determines the importance of each moment condition in the estimation objective. While GMM
203  estimators are consistent and normal under *any* choice of positive definite $\widehat{\mathbf{W}}_T$, the selection of $\widehat{\mathbf{W}}_T$
204  influences their efficiency.

205  **Two-step GMM estimator.**  We adopt the two-step GMM procedure as described in Newey and
206  McFadden [1994]. First, we compute the one-step estimator $\hat{\theta}_T^{(\text{os})}, \hat{\eta}_T^{(\text{os})}$ using an identity weight
207  matrix $\widehat{\mathbf{W}}_T = \mathbf{I}$. Then, we estimate the optimal weight matrix as:

$$
\widehat{\Omega}_T(\hat{\theta}_T^{(\text{os})}, \hat{\eta}_T^{(\text{os})}) = \left[ \frac{1}{T} \sum_{t=1}^{T} g_t(\hat{\theta}_T^{(\text{os})}, \hat{\eta}_T^{(\text{os})}) g_t(\hat{\theta}_T^{(\text{os})}, \hat{\eta}_T^{(\text{os})})^{\top} \right],
\tag{4}
$$

208  and set

$$
\widehat{\mathbf{W}}_T = \left[ \widehat{\Omega}_T(\hat{\theta}_T^{(\text{os})}, \hat{\eta}_T^{(\text{os})}) \right]^{-1}.
\tag{5}
$$

209  This optimal weighting has the interpretation as the inverse empirical covariance of the moment
210  conditions on the one-step estimate. We then compute the final two-step estimator by minimizing
211  $\widehat{Q}_T(\theta)$ with this updated weighting matrix. This choice of $\widehat{\mathbf{W}}_T$ yields an asymptotically efficient
212  estimator under standard GMM regularity conditions.

213  The adoption of two-step GMM is a critical component of our proposed estimation framework.
214  Indeed, in the first-step estimates, the synthetic and proxy data will have no impact on the estimate
215  of $\theta$ because they never appear in the moment conditions concerning $\theta$. In the second stage though,
216  the weight matrix $\widehat{\mathbf{W}}_T$ accounts for the covariance between moment conditions, where off-diagonal
217  terms in the matrix allow moments for the auxiliary data sources to influence the estimation of $\theta$.

## 4.4 Consistency and Asymptotic Inference

We now present results on the consistency and asymptotic behavior of our GMM estimators.

**Proposition 1.** *Our estimate $\hat{\theta}_T$ (as defined in Equation 3) is consistent and asymptotically normal. It converges in distribution as*

$$\sqrt{T}((\hat{\theta}_T', \hat{\eta}_T')' - (\theta', \eta')') \xrightarrow{d} \mathcal{N}(0, V)$$

*where the covariance $V$ is given by*

$$V = \left(G(\theta,\eta)^T \widehat{\mathbf{W}} G(\theta,\eta)\right)^{-1} G(\theta,\eta)^T \widehat{\mathbf{W}} F \widehat{\mathbf{W}} G(\theta,\eta) \left(G(\theta,\eta)^T \widehat{\mathbf{W}} G(\theta,\eta)\right)^{-1},$$

*and where $G(\theta,\eta)$ is the Jacobian of the population moments at the ground truth parameter values $\theta, \eta$.*

For optimal weight matrix 5, this simplifies to $V = (G(\theta,\eta)^T F^{-1} G(\theta,\eta))^{-1}$. These are standard results on GMM estimators, which follow by straightforwardly applying the results in Hansen [1982]. We remark that these asymptotic results require a set of conditions on the sample moments, which are slightly nuanced in this setting with multiple sources of information. We discuss these conditions and prove that they are satisfied in Appendix A for the setting of proxy and synthetic samples. Given this asymptotic behavior, we can derive valid confidence intervals for our parameter estimates.

## 4.5 Why does synthetic data improve performance?

To understand where the benefits arise from incorporating the proxy and synthetic data into our GMM estimator, we analyze the interaction between our moment conditions. Note that the functions $\psi$ are often referred to as "residuals" in the GMM literature; since $\psi(\theta)$ should be zero in-expectation, deviations from zero are interpretable as a kind of residual. The key intuition is that synthetic data will improve performance when the synthetic-data residuals are predictive of the real-data residuals.

First, we note that if the synthetic data were perfectly simulated, $X$ and $Y$ would be perfectly recovered from the unlabeled text $T$. With ground truth $X, Y$, we can perfectly recover the residual terms. In settings where we have good but imperfect simulations, $\hat{X}, \hat{Y}$ and $\tilde{X}, \tilde{Y}$ are highly correlated with the errors in the true data, and we can approximately estimate the real-data residuals with the synthetic data. Within our GMM-based approach, this is all handled implicitly in our two-step estimation procedure. During the first estimation step, each set of parameters (e.g., defined on the observed, proxy, and synthetic data) is independently identified since the initial weighting is an identity matrix. The key insight is that, during the second estimation step, the weighting matrix $\widehat{\mathbf{W}}$, which is the inverse of the moment covariance matrix, captures the interactions between the observed residual terms and the residuals from the synthetic data in our GMM objective.

Partitioning the moments into observed data residuals $m_t(\theta)$ and synthetic data residuals $h_t(\eta)$, we derive an explicit formula for the asymptotic variance of $\sqrt{T}(\hat{\theta}_T - \theta)$ in Appendix C. We find two important conclusions. First, when these residuals are independent of the observed data, the formula reduces to the optimal variance based only on the fully observed data. That is, in the worst case where synthetic data is completely uninformative, including it does not hurt (at least asymptotically). Second, when the real and synthetic residuals are correlated (as we would hope), we derive a lower bound on the variance which is proportional to the residual variance in a regression of the observed data residuals on the span of the synthetic data residuals. This bound is minimized by choosing moments that span the conditional expectation of the observed data residuals given $T_i$, a sufficient condition for which is that the conditional distribution of $\hat{X}, \hat{Y}$ or $\tilde{X}, \tilde{Y}$ given $T$ equals the conditional distribution of $X, Y$.

# 5 How to Apply a Debiasing-based Approach

In addition to proposing our GMM-based estimator, we consider how existing debiasing-based methods, such as PPI++ [Angelopoulos et al., 2023b], might be adapted to our setting. These methods have been well-studied in the context of predicted outcomes and, more recently, predicted covariates. However, it is not immediately clear how to aggregate multiple sources of information (i.e., proxy

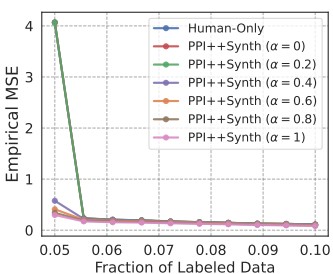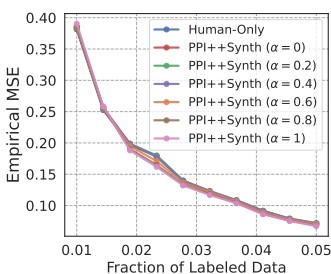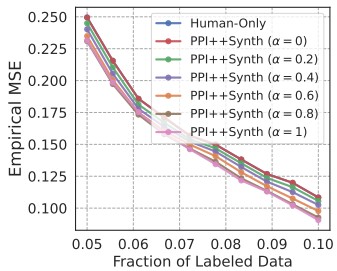

Figure 1: Grid search of the proposed debiasing-based approach (PPI++Synth) across different $\alpha$ values (on 1PP, Hedging, and Stance estimation tasks (from left to right)). We can observe that the optimal $\alpha$ value amongst the ones searched is defaulted to 1 in all cases, which is equivalent to collapsing to fully using the proxy data. Results are averaged over 200 trials.

data and synthetic data) in this setup. Perhaps the most general approach is given by RePPI [Ji et al., 2025], which predicts the optimal loss through fitting an arbitrary model that maps the proxy and synthetic loss to the real loss. In order to limit the number of parameters, we examine a natural instantiation of this, where the model is a convex combination.

**Proposition 2.** *The adapted, debiasing-based loss objective with multiple predicted covariates and outcomes is given by*

$$L^{PP}(\theta) := \frac{1}{N} \sum_{i=1}^{N} [(1-\alpha) \cdot l_\theta(\tilde{X}_i, \tilde{Y}_i) + \alpha \cdot l_\theta(\hat{X}_i, \hat{Y}_i)] \tag{6}$$

$$+ \frac{1}{n} \sum_{i=1}^{n} (l_\theta(X_i, Y_i) - [(1-\alpha) \cdot l_\theta(\tilde{X}_i, \tilde{Y}_i) + \alpha \cdot l_\theta(\hat{X}_i, \hat{Y}_i)]). \tag{7}$$

*where the estimate retains asymptotic normality conditions (see Appendix for the proof and algorithm details).*

Importantly, note that the addition of this hyperparameter $\alpha$ adds increased complexity, and techniques such as cross-fitting must be used to select it in a statistically valid fashion. We refer to the estimator with $\alpha = 1$ as **PPI++Proxy**, as the synthetic terms vanish, yielding an estimator that combines real and proxy data. We refer to the estimator with tunable $\alpha \in [0, 1]$ as **PPI++Synth**, which combines real, proxy, and synthetic data. We note that our implementation builds on PPI++, retaining all additional benefits, such as power tuning, over the standard PPI estimator.

## 6 Experimental Results

We evaluate the finite-sample performance of our proposed estimators (GMM-Synth and GMM-Proxy) as well as the adapted debiasing-based estimators (PPI++Synth and PPI++Proxy) in the following setup.

**Datasets and Experimental Setup.** We focus on the small-data regime, where the need for additional data sources is especially well-motivated. In particular, we consider settings where the practitioner has a corpus of unlabeled text and only a small set of human-annotated samples (e.g., ground-truth covariates and labels derived from the text). We evaluate our framework in four different computational social science tasks, each involving a regression coefficient as the target quantity. In the first two tasks, we use texts from online requests posted on Stack Exchange and Wikipedia [Danescu-Niculescu-Mizil et al., 2013] to estimate how certain linguistic features affect perceived politeness; specifically, the use of first-person plural pronouns and the presence of hedging markers (i.e., expressions of uncertainty). The third task examines the effect of affirming linguistic devices on media stance toward global warming (i.e., whether the news headline supports or rejects climate change) using a corpus of climate-related news headlines [Hmielowski et al., 2014]. Finally, in the fourth task, we analyze congressional bills texts [Adler and Wilkerson, 2011] to estimate the effect of a legislator's DW-Nominate measure [Lewis et al., 2024] of ideology on the type of bill (whether

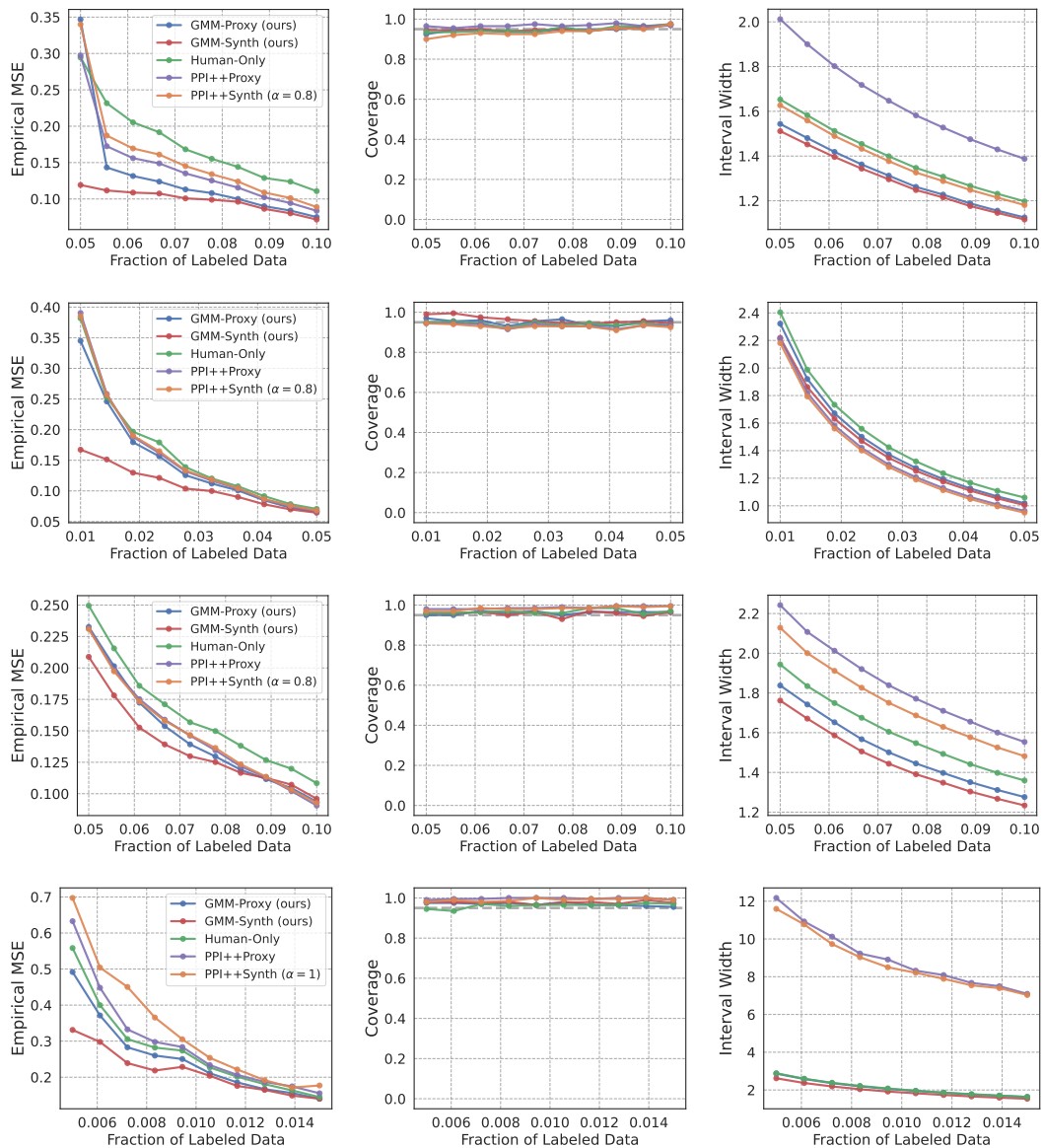

Figure 2: **Main Results**. We observe large reductions in MSE, especially in very low-label regimes. Each row corresponds to a task (i.e., 1pp, Hedging, Stance (from top to bottom)); each column corresponds to a metric (i.e., MSE, coverage, confidence interval width (from left to right)). Note that when the best performing PPI++Synth is equivalent to PPI++Proxy, we report the second-best performing PPI++Synth method ($\alpha = 0.8$ for these tasks). Results are averaged over 200 trials.

the bill pertains to macroeconomy). In all the tasks, the target quantity is the regression coefficient corresponding to the explanatory variable of interest.

To evaluate our framework, we use GPT-4o [Hurst et al., 2024] to generate proxy and synthetic data, without any task-specific fine-tuning, i.e., using the LLM out of the box. We report the empirical mean-squared error (MSE), coverage at level $\alpha = 0.05$, confidence interval width, and effective sample size across all tasks. The effective sample size $n_{\text{effective}}$ denotes the number of human-labeled samples needed for the classical estimator $\hat{\theta}^{\text{human}}$ to match the MSE of the method's estimate $\hat{\theta}^{\text{method}}$. In other words, it quantifies how many human annotations the method effectively saves while maintaining equivalent accuracy. We defer the results and discussion for effective sample size results to the Appendix.

**Key Observations.** We begin by presenting our main results. In Figure 2, we evaluate the performance of our GMM-based estimators: GMM-Proxy and GMM-Synth. Across all studied tasks, we observe both methods consistently outperform only leveraging ground-truth human-annotated samples (Human-only), yielding improvements in both point estimation (MSE) and inference (tighter intervals while retaining proper coverage). As expected, we observe that benefits are especially pronounced in low-label regimes, which aligns with the motivating use case of our framework. On several tasks in low-label regimes, we observe large reductions in MSE (more than 50% reductions) compared to only using the human-labeled samples. We note that across all settings, the proxy data and synthetic data alone yield greatly biased estimates (see Appendix). However, the specific structure in how we combine these data sources with human-labeled data enables better estimation of the target parameter. See Section 4.5 for a deeper analysis of how this interaction improves performance.

Next, we turn to analyzing the results of our adapted debiasing-based estimators, which we refer to as PPI++Proxy and PPI++Synth for convenience. Note that in the implementation of our debiasing-based estimators, we leverage PPI++ [Angelopoulos et al., 2023b], which further includes benefits of power tuning. We empirically find that PPI++Synth often underperforms in regimes, where the sample size of labeled data is small, due to cross-fitting restricting the sample size even further. As an upper bound, we conduct a grid search over different possible $\alpha$ values *without* cross-fitting. Note, this is not a valid solution in the setup, since this requires cheating in hyperparameter selection. In Figure 1, we empirically find that although this oracle incorporates proxy data effectively, introducing the synthetic data does not yield further performance improvement. We can clearly observe that the optimal $\alpha$ is 1 in all cases, which is equivalent to only utilizing information from the proxy data terms (i.e., ignoring the synthetic data terms completely). In Figure 2, we observe that although both methods retain reasonable coverage, we see that they underperform the GMM-based estimators, resulting in larger MSE and mostly wider intervals.

# 7 Discussion

In this work, we introduce a principled framework for incorporating fully synthetic samples into downstream statistical analyses. We provide practical guidance for constructing synthetic samples in ways that support valid inference, and propose a new estimator based on generalized method of moments (GMM) estimation, where the key intuition is that synthetic data will improve performance when the synthetic-data residuals are predictive of the real-data residuals. Across the studied regression tasks, we indeed observe a large degree of improvements in estimation, especially in very low-label regimes. More broadly, this work takes a first step toward understanding how imperfect synthetic data can systematically be leveraged to support valid inference. As the usage and future promise of LLMs continue to grow, so too will the complexity of pipelines that incorporate their outputs. Our framework provides one route towards easily-extensible estimation methods that can flexibly incorporate growing variety and quality in synthetic data sources.

**Limitations.** A potential limitation of our framework is its reliance on the quality of the generative model (e.g., an LLM). As with other debiasing approaches, very poor-quality synthetic data would yield little-to-no benefits in statistical efficiency. Moreover, our theoretical guarantees, like those of debiasing methods, hold asymptotically and thus may fail to hold in extremely low-data regimes, potentially leading to undercoverage of the target parameter. Furthermore, we note that our framework assumes a specific sampling procedure in which each synthetic sample is conditioned on a real sample. In cases where synthetic data is generated differently—such as via zero-shot prompting without conditioning—our framework may not apply.

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

# A Conditions for Consistency and Asymptotic Normality

We provide a discussion about the necessary conditions for a GMM estimator to be consistent and asymptotically normal, showing that these conditions are indeed met for our augmented GMM.

As mentioned in the construction of our estimator, we define one moment condition for each parameter on the observed data $D$. We also define two moments for each parameter on the proxy and synthetic data. This leads to an overidentified system, with more moments than parameters, ensuring that the target parameter is identifiable.

Next, we establish a few conditions for valid asymptotic properties of our GMM estimator, specifically about the convergence and distributions of the sample moments. First, we require that all of our moments converge to their expectation, or that

$$\frac{1}{n} \sum_{i=1}^{n} \psi^{(j)} \to \mathbb{E}[\psi^{(j)}].$$

Next, they must also obey the central limit theorem, or that

$$\sqrt{n} \left( \frac{1}{n} \sum_{i=1}^{n} \psi^{(j)} \right) \xrightarrow{d} \mathcal{N}(0, F),$$

where $F$ is some finite covariance matrix of all the moments.

Under these standard regularity conditions on the moment functions $\psi$ [Newey and McFadden, 1994], these conditions are immediately satisfied for the moments defined on observed data, as each observation of the moments are independent. The same holds for the moments defined on proxy data, since $\hat{X}, \hat{Y}$ are functions of independent inputs $T$, and are therefore also independent across observations. The case of synthetic data is slightly more nuanced, but we show that the required conditions still hold, through the following lemma.

**Lemma 1.** *Let $\{\phi\}_{j=1}^{m}$ represent our moments defined on synthetic observations. Then, they are i.i.d., and consequently*

$$\frac{1}{m} \sum_{j=1}^{m} \phi_j \; \to \; \mathbb{E}[\phi_j] \quad and \quad \sqrt{m} \left( \frac{1}{m} \sum_{j=1}^{m} \phi_j \right) \; \xrightarrow{d} \; \mathcal{N}(0, \sigma(\phi)),$$

*where $\sigma(\phi)$ is the variance matrix of $\phi$.*

*Proof.* We begin by noting that the unlabeled texts $\{T_j\}_{j=1}^{m}$ are drawn i.i.d. from the marginal distribution $\mathcal{D}_T$. For each $T_j$, a synthetic text $\tilde{T}_j$ is generated by a generative model (i.e., an LLM), which uses independent randomness for each call. The model is conditioned only on an individual sample $(T_j, X_j)$ if $j$ is labeled or $(T_j, \hat{X}_j)$ otherwise. Since the generative process for each $T_j$ is independent and the mapping $\tilde{T}_j \mapsto (\tilde{X}_j, \tilde{Y}_j)$ is applied identically to each sample, the resulting pairs $(\tilde{X}_j, \tilde{Y}_j)$ are also i.i.d. As these pairs are drawn i.i.d., then these conditions are met via the central limit theorem. $\qquad\square$

This result shows that the required conditions on the sample moments hold in our setting of proxy and synthetic samples; under the regularity conditions of Newey and McFadden [1994] Theorem 3.2, one immediately obtains Proposition 1 on the asymptotic behavior of our GMM estimator.

# B Moment Conditions

We provide a concrete example of our moment construction for the case of generalized linear models (GLMs) in two-dimensions.

 **B.1   Example 1. Generalized Linear Models**

459   Recall that the standard GLM formulation optimizes the objective function,

$$\ell_\theta(x, y) = -yx^T\theta + f(x^T\theta),$$

460   where $f$ is a function that is convex and infinitely differentiable. We remark that this recovers the
461   setting of logistic regression when $f(z) = \log(1 + \exp(z))$. Let us assume a two-dimensional setting
462   for illustration. This translates to the population moment conditions of

$$\mathbb{E}\left[X_1\left(Y - \frac{\partial f}{\partial\theta_1}(X^T\theta^*)\right)\right] = 0, \quad \mathbb{E}\left[X_2\left(Y - \frac{\partial f}{\partial\theta_2}(X^T\theta^*)\right)\right] = 0$$

463   We have similar moments for proxy and synthetic data, where we use parameters $\eta = (\eta^{(1)}, \eta^{(2)})$,
464   which are also two-dimensional. Within our GMM framework, we construct the following set of
465   moment conditions across the observed, proxy, and synthetic data.

$$g_t(\theta, \eta) = \begin{bmatrix} s_t \\ s_t \\ s_t \\ s_t \\ s_t \\ s_t \\ 1 \\ 1 \\ 1 \\ 1 \end{bmatrix} \odot \begin{bmatrix} X_{t,1}(Y_t - \frac{\partial f}{\partial\theta_1}(X_t^T\theta)) \\ X_{t,2}(Y_t - \frac{\partial f}{\partial\theta_2}(X_t^T\theta)) \\ \hat{X}_{t,1}(\hat{Y}_t - \frac{\partial f}{\partial\eta_1^{(1)}}(\hat{X}_t^T\eta^{(1)})) \\ \hat{X}_{t,2}(\hat{Y}_t - \frac{\partial f}{\partial\eta_2^{(1)}}(\hat{X}_t^T\eta^{(1)})) \\ \tilde{X}_{t,1}(\tilde{Y}_t - \frac{\partial f}{\partial\eta_1^{(2)}}(\tilde{X}_t^T\eta^{(2)})) \\ \tilde{X}_{t,2}(\tilde{Y}_t - \frac{\partial f}{\partial\eta_2^{(2)}}(\tilde{X}_t^T\eta^{(2)})) \\ \hat{X}_{t,1}(\hat{Y}_t - \frac{\partial f}{\partial\eta_1^{(1)}}(\hat{X}_t^T\eta^{(1)})) \\ \hat{X}_{t,2}(\hat{Y}_t - \frac{\partial f}{\partial\eta_2^{(1)}}(\hat{X}_t^T\eta^{(1)})) \\ \tilde{X}_{t,1}(\tilde{Y}_t - \frac{\partial f}{\partial\eta_1^{(2)}}(\tilde{X}_t^T\eta^{(2)})) \\ \tilde{X}_{t,2}(\tilde{Y}_t - \frac{\partial f}{\partial\eta_2^{(2)}}(\tilde{X}_t^T\eta^{(2)})) \end{bmatrix}$$

466   # C   Partitioned GMM Asymptotic Variance

467   We now derive the asymptotic variance of our GMM estimator for specifically the target parameter
468   $\hat{\theta}_T$.

469   *

470   *Proof.* With the optimal choice of weight matrix for the full GMM estimation problem, the asymptotic
471   variance of the vector $(\hat{\theta}, \hat{\eta})$ converges to $(G^T F^{-1} G)^{-1}$. To obtain the variance for $\hat{\theta}$ specifically,
472   partition the moments into $g_t(\theta, \eta) = (m_t(\theta)', h_t(\eta)')'$, where $m_t(\theta) = S_t \odot \psi(\theta)$, and

$$h_t(\eta) = \begin{bmatrix} S_t \\ S_t \\ \vdots \\ S_t \\ 1 \\ \vdots \\ 1 \end{bmatrix} \odot \begin{bmatrix} \psi(\eta^{(1)}) \\ \vdots \\ \psi(\eta^{(M)}) \\ \psi(\eta^{(1)}) \\ \vdots \\ \psi(\eta^{(M)}) \end{bmatrix}$$

473   Given this partitioning, we can express

$$G(\theta, \eta) = \begin{bmatrix} \frac{d\mathbb{E}[m(\theta)]}{d\theta} & 0 \\ 0 & \frac{d\mathbb{E}[h(\eta)]}{d\eta} \end{bmatrix}$$

$$F = \begin{bmatrix} \mathbb{E}[m_t(\theta)m_t(\theta)'] & \mathbb{E}[m_t(\theta)h_t(\eta)'] \\ \mathbb{E}[h_t(\eta)m_t(\theta)'] & \mathbb{E}[h_t(\theta)h_t(\theta)'] \end{bmatrix}$$

474   By the partitioned inverse formula, we can express $F^{-1}$ as

$$\begin{bmatrix} A & B \\ B^\top & D \end{bmatrix}$$

475   where the upper left block $A$ is

$$(\mathbb{E}[m_t(\theta)m_t(\theta)'] - \mathbb{E}[m_t(\theta)h_t(\eta)']\mathbb{E}[h_t(\theta)h_t(\theta)']^{-1}\mathbb{E}[h_t(\eta)m_t(\theta)'])^{-1}$$

476   This term can be interpreted as the inverse of the asymptotic residual variance of a regression of
477   $m_t(\theta)$ on the span of the vector $h_t(\eta)$.

478   The lower right block $D$ is, symmetrically, the asymptotic residual variance of a regression of $h_t(\theta)$
479   on the span of the vector $m_t(\eta)$:

$$(\mathbb{E}[h_t(\theta)h_t(\theta)'] - \mathbb{E}[h_t(\theta)m_t(\eta)']\mathbb{E}[m_t(\theta)m_t(\theta)']^{-1}\mathbb{E}[m_t(\eta)h_t(\theta)'])^{-1}$$

480   Finally, the off-diagonal term multiplies $A$ by the coefficient in a regression of $m$ on $h$:

$$B = -A\mathbb{E}[m_t(\theta)h_t(\eta)']\mathbb{E}[h_t(\theta)h_t(\theta)']^{-1}$$

481   For the full variance,

$$G^\top F^{-1} G = \begin{bmatrix} \frac{d\mathbb{E}[m(\theta)]}{d\theta'}A\frac{d\mathbb{E}[m(\theta)]}{d\theta} & \frac{d\mathbb{E}[m(\theta)]}{d\theta'}B\frac{d\mathbb{E}[h(\eta)]}{d\eta} \\ \frac{d\mathbb{E}[h(\eta)]}{d\eta'}B^\top\frac{d\mathbb{E}[m(\theta)]}{d\theta} & \frac{d\mathbb{E}[h(\eta)]}{d\eta'}D\frac{d\mathbb{E}[h(\eta)]}{d\eta} \end{bmatrix}$$

Applying the partitioned inverse formula again, the upper left block of $(G^\top F^{-1} G)^{-1}$, which gives
exactly the asymptotic variance of $\sqrt{T}(\hat{\theta}_T - \theta)$, is equal to

$$(\frac{d\mathbb{E}[m(\theta)]}{d\theta'}A\frac{d\mathbb{E}[m(\theta)]}{d\theta} - \frac{d\mathbb{E}[h(\eta)]}{d\eta'}B^\top\frac{d\mathbb{E}[m(\theta)]}{d\theta}(\frac{d\mathbb{E}[h(\eta)]}{d\eta'}D\frac{d\mathbb{E}[h(\eta)]}{d\eta})^{-1}\frac{d\mathbb{E}[m(\theta)]}{d\theta'}B\frac{d\mathbb{E}[h(\eta)]}{d\eta})^{-1}$$

482   This can be interpreted similarly as the asymptotic variance of the residual prediction error from a
483   regression of $A^{-1/2}\frac{dm(\theta)}{d\theta}$ onto the span of a weighted linear combination of terms in $\frac{dh(\eta)}{d\eta}$.   $\square$

We remark that a lower bound on the total variance is given by $(\frac{d\mathbb{E}[m(\theta)]}{d\theta'}A\frac{d\mathbb{E}[m(\theta)]}{d\theta})^{-1}$, which is
minimized when $A$ is maximized. Among choices of moment functions $h_t(\eta)$ that depend solely on
$T_t$, $A$ is maximized in the positive semi-definite order when the span of $h_t(\eta)$ contains $\mathbb{E}[m(\theta)|T_t]$. A
sufficient but not necessary condition for this is that for some $j \in 1 \ldots M$, the conditional moments
of the simulation are identical to those of the real data:

$$E[\psi(\eta_j)|T_i] = E[\psi(\theta)|T_i]$$

484   This calibration condition is satisfied when the conditional distribution of the simulated data given
485   $T$ equals that of the real data, which is a natural simulation target, though not required for valid
486   inference.

# D   Baseline Details

## D.1   PPI++Proxy and PPI++Synth Implementation

489   We now present a discussion on our adapted debiasing-based approach from Proposition 2.

---

**Algorithm 1** Cross-Fitting for PPI⁺⁺Synth

---

**Require:**
1: Labeled data $\mathcal{D} = \{(T_i, X_i, Y_i)\}_{i=1}^n$,
2: Proxy data $\widehat{\mathcal{D}} = \{(T_j, \widehat{X}_j, \widehat{Y}_j)\}_{j=1}^{n+m}$,
3: Synthetic data $\widetilde{\mathcal{D}} = \{(\widetilde{T}_j, \widetilde{X}_j, \widetilde{Y}_j)\}_{j=1}^{n+m}$,
4: K folds
**Ensure:** Debiased estimate $\hat{\theta}_{\text{CF}}$
5: Split $\mathcal{D}$ into folds $\{\mathcal{I}_1, \ldots, \mathcal{I}_K\}$
6:
7: **for** $k = 1, \ldots, K$ **do**
8:      define train-fold $\mathcal{I}_{\text{train}} = \bigcup_{r \neq k} \mathcal{I}_r$
9:      $\hat{\theta}_1^{-k} \leftarrow \arg\min_\theta L_{\text{PP}}^{-k}(\theta; 0)$            $\triangleright$ (1) initial fit on train-fold
10:
11:      $\hat{\alpha}^{-k} \leftarrow \arg\min_{\alpha \in [0,1]} L_{\text{PP}}^{-k}(\hat{\theta}_1^{-k}; \alpha)$     $\triangleright$ (2) select mixture weight $\alpha$ on train-fold)
12:
13:      $\hat{\theta}^k \leftarrow \arg\min_\theta L_{\text{PP}}^k(\theta; \hat{\alpha}^{-k})$       $\triangleright$ (3) final fit on held-out fold with chosen $\alpha$)
14:
15: **end for**
16: **return** $\hat{\theta}_{\text{CF}} = \dfrac{1}{K} \sum_{k=1}^K \hat{\theta}^k$

---

### D.1.1 Asymptotic Normality

First, it is relatively straightforward to show that this is an unbiased estimate of the true objective.

$$\begin{aligned}
\mathbb{E}[L^{PP}(\theta)] &= (1-\alpha) \cdot \mathbb{E}[l_\theta(\tilde{X}, \tilde{Y})] + \alpha \cdot \mathbb{E}[l_\theta(\hat{X}, \hat{Y})] \\
&\quad + \mathbb{E}[l_\theta(X, Y)] - \mathbb{E}[(1-\alpha) \cdot l_\theta(\tilde{X}, \tilde{Y})] - \alpha \cdot \mathbb{E}[l_\theta(\hat{X}, \hat{Y})])] \\
&= \mathbb{E}[\ell_\theta(X, Y)].
\end{aligned}$$

Note that this holds for any choice of the hyperparameter $\alpha$.

Under the same assumptions as in the PPI++ paper [Angelopoulos et al., 2023b] (e.g., that $\frac{n}{n+m} \to c$ for some constant $c$ and, in the case of generalized linear models, the Hessian is non-singular, we perform their same approach to power tuning), we recover the asymptotic normality guarantees of the parameter estimate (as in Corollary 1 from Angelopoulos et al. [2023b]).

### D.1.2 Hyperparameter Selection via Cross-fitting

The added complexity from these modified debiasing-based approaches arises from the hyperparameter $\alpha$. We now discuss an approach for selecting $\alpha$ by performing cross-fitting. As previously mentioned, we can treat $\alpha$ as a simple version of RePPI [Ji et al., 2025] where we fit a convex combination of proxy and synthetic losses.

Namely, we partition our available data into two splits. We select $\alpha$ on one fold by minimizing:

$$\arg\min_{\alpha \in [0,1]} L^{PP}(\theta_1),$$

where $\theta_1$ is defined as the solution to the naive minimzation of $\mathbb{E}[\ell_\theta(X, Y)]$ on the same split. This essentially captures picking the $\alpha$ that best combines the proxy and synthetic losses to best mimic the behavior of the standard loss function.

We then take this optimal $\alpha$ and use it to produce a parameter estimate on the held-out fold. We aggregate these estimates as is standard in cross-fitting approaches. We outline this process in Algorithm 1.

# E  Experimental Details

## E.1  Prompt Texts

We present the full text prompts that were used to generate proxy covariates and labels (for the proxy data) and synthetic data. Note that the prompts used to extract covariates and labels from the synthetic text are identical to those used for the proxy data.

---

**Proxy Data Generation Prompts**

**Politeness (First Plural Pronouns) - Covariates:**
Does the following text contain first person plural pronouns (e.g., we, us, our, ourselves)? Output either yes or no.
Text: """
{content}
"""
**Answer:**

**Politeness (First Plural Pronouns) - Labels:**
Is the following text polite? Output either A or B. Output a letter only.
A) Polite
B) Impolite
Text: """
{content}
"""
**Answer:**

**Politeness (Hedging) - Covariates:**
Does the following text contain hedging devices—expressions that indicate uncertainty, caution, or a lack of full commitment to a claim (e.g., may, might, could, would, possibly, probably, perhaps, apparently, suggest, indicate, seem, appear, it is likely that, it seems that)? Respond with yes or no only.
Text: """
{content}
"""
**Answer:**

**Politeness (Hedging) - Labels:**
Is the following text polite? Output either A or B. Output a letter only.
A) Polite
B) Impolite
Text: """
{content}
"""
**Answer:**

**Stance Dataset - Covariates:**
Does the following text contain any affirmative device words? Output either yes or no.
Text: """
{content}
"""
**Answer:**

---

**Stance Dataset - Labels:**
A statement can agree, be neutral, or disagree with the statement: "Climate change/global warming is a serious concern". Classify the following statement into one of the three categories. Output either A, B, or C. Output a letter only.
A) Agree
B) Neutral
C) Disagree
Statement: """
{content}
"""
**Answer:**

**Congressional Bills Dataset - Covariates:**
You are a political scientist familiar with the U.S. Congress and the DW-NOMINATE scoring system, which places legislators and legislation on a left-right ideological spectrum ranging approximately from -1 (most liberal) to +1 (most conservative). Below is the text of a proposed bill. Based on the policy content, language, and framing of the bill, estimate the DW-NOMINATE score that best represents its ideological position. Output a single nonzero float between -1 and +1 representing the estimated DW-NOMINATE score of the bill.
Bill: """
{content}
"""
**Answer:**

**Congressional Bills Dataset - Labels:**
Does the following text relate to the economy? Output either true or false.
Text: """
{content}
"""
**Label:**

515

---

**Synthetic Data Generation Prompts**

**Politeness (First Plural Pronouns)**
Consider texts taken from user requests on Stack Exchange or Wikipedia. Each text is labeled as either polite or impolite, and either contains or does not contain first-person plural pronouns. Below is an example that {x}:
**Example:** """
{example}
"""
Now, generate a new example of a request that also {x}.

**Politeness (Hedging)**
Consider texts taken from user requests on Stack Exchange or Wikipedia. Each text can be labeled as either polite or impolite, and as either containing a hedging device or not containing one. Hedging devices are expressions that indicate uncertainty, caution, or a lack of full commitment to a claim (e.g., may, might, could, would, possibly, probably, perhaps, apparently, suggest, indicate, etc.). Below is an example that {x}:
**Example:** """
{example}
"""
Now, generate a new example of a request that also {x}.

516

## Synthetic Data Generation Prompts (continued)

**Stance**

Consider news headlines that take a stance — agree, disagree, or neutral — on the statement:
"Climate change/global warming is a serious concern."
Each headline also either contains or does not contain an affirmative device.
Below is an example of a headline.
**Example:** """
{example}
"""
Affirmative device: {x}
Now, generate a new news headline about global warming that also {x}.

**Congressional Bills Data**

You are a political language model trained to generate realistic examples of U.S. congressional
bills. Each bill is labeled as either "related to the economy" or "not related to the economy",
and is associated with a DW-NOMINATE score representing ideological position (ranging
from $-1$ liberal to $+1$ conservative).
**Example:**
Bill Text: """
{example}
"""
DW-NOMINATE Score: {dw_nominate_score}
Now, generate a new example of a bill that also has a DW-NOMINATE score of
{dw_nominate_score}. Output only the new bill text: """