# OpenReview forum: "Drawing Reliable Conclusions with Imperfect Synthetic Data"
_NeurIPS.cc/2025/Workshop/Reliable_ML — NeurIPS 2025 - Reliable ML Workshop_

### Official Review · Reviewer_rGoS · 2025-09-18
**Promising Framework for Inference with Synthetic Data via GMM**

**Rating:** 7
**Confidence:** 3

**Review:**

Summary:

The paper proposes an estimator based on generalised method of moments (GMM) to incorporate synthetic data along with real data to draw reliable statistical conclusion in downstream tasks. The authors show that moment residuals of synthetic data and that of the real data in the weighting matrix can improve the estimates of the target parameter. Moreover, they show that even if the synthetic data is completely uninformative, including it does not deteriorate the performance in downstream tasks.

Strengths:

1. The research problem is relevant and timely. As LLMs are increasingly being used to generate synthetic data including simulating human responses, determining how well these simulations resemble the real world is an important question.
2. The theoretical results are presented rigorously, and the empirical evaluations are detailed and clearly reported.

Weakness:
1. More experiments using the proposed method in other domains beyond text could add substantial value to the paper. That being said, the work is quite detailed and refined for a workshop submission.

---

### Official Review · Reviewer_NbHP · 2025-09-20

**Rating:** 8
**Confidence:** 4

**Review:**

The paper considers the problem of performing synthetic data-assisted estimation. Specifically, it is assumed that we have access to a small number of real data from a source, and we want to augment the dataset using synthetic data. The paper proposes a method relying on the Generalized Method of Moments to fit a machine learning model to an underlying population, in a way that will result in the model to be able to generate synthetic data that can be used to in addition to real data.

This is a good paper that addresses an important topic. Both theoretical and empirical evidence is offered to support the method's performance, which makes this a well-rounded submission in my opinion.